# Network Pharmacological Analysis and Experimental Validation of the Effect of *Smilacis Glabrae Rhixoma* on Gastrointestinal Motility Disorder

**DOI:** 10.3390/plants12071509

**Published:** 2023-03-30

**Authors:** Na-Ri Choi, Kangwook Lee, Mujin Seo, Seok-Jae Ko, Woo-Gyun Choi, Sang-Chan Kim, Jinsung Kim, Jae-Woo Park, Byung-Joo Kim

**Affiliations:** 1Division of Longevity and Biofunctional Medicine, School of Korean Medicine, Pusan National University, Yangsan 50612, Republic of Korea; nariring@gmail.com (N.-R.C.); seorafino1012@gmail.com (M.S.); ak0510@hanmail.net (W.-G.C.); 2Department of Clinical Korean Medicine, Graduate School of Kyung Hee University, Seoul 02447, Republic of Korea; dkwkgo3@naver.com (K.L.); kokokoko119@daum.net (S.-J.K.); oridoc@khu.ac.kr (J.K.); 3Department of Gastroenterology, College of Korean Medicine, Kyung Hee University, Seoul 02447, Republic of Korea; 4College of Oriental Medicine, Daegu Haany University, Gyeongsan 38610, Republic of Korea; sckim@dhu.ac.kr

**Keywords:** gastrointestinal motility disorder, *Smilacis Glabrae Rhixoma*, network pharmacology, gastric emptying, intestinal transit rate

## Abstract

Gastrointestinal motility disorder (GMD) is a disease that causes digestive problems due to inhibition of the movement of the gastrointestinal tract and is one of the diseases that reduce the quality of life of modern people. *Smilacis Glabrae Rhixoma* (SGR) is a traditional herbal medicine for many diseases and is sometimes prescribed to improve digestion. As a network pharmacological approach, we searched the TCMSP database for SGR, reviewed its constituents and target genes, and analyzed its relevance to gastrointestinal motility disorder. The effects of the SGR extract on the pacemaker activity in interstitial cells of Cajal (ICC) and gastric emptying were investigated. In addition, using the GMD mouse model through acetic acid (AA), we investigated the locomotor effect of SGR on the intestinal transit rate (ITR). As a result of network pharmacology analysis, 56 compounds out of 74 candidate compounds of SGR have targets, the number of targets is 390 targets, and there are 904 combinations. Seventeen compounds of SGR were related to GMD, and as a result of comparing the related genes with the GMD-related genes, 17 genes (active only) corresponded to both. When looking at the relationship network between GMD and SGR, it was confirmed that quercetin, resveratrol, SCN5A, TNF, and FOS were most closely related to GMD. In addition, the SGR extract regulated the pacemaker activity in ICC and recovered the delayed gastric emptying. As a result of feeding the SGR extract to AA-induced GMD mice, it was confirmed that the ITR decreased by AA was restored by the SGR extract. Through network pharmacology, it was confirmed that quercetin, resveratrol, SCN5A, TNF, and FOS were related to GMD in SGR, and these were closely related to intestinal motility. Based on these results, it is suggested that SGR in GMD restores digestion through the recovery of intestinal motility.

## 1. Introduction

Gastrointestinal motility disorder (GMD) is a disease in which digestive function is impaired due to impaired motility of the gastrointestinal tract and can occur throughout the intestine, including the esophagus and colon. These gastrointestinal motility disorders are accompanied by a variety of chronic symptoms, including vomiting and nausea [1]. Many people experience discomfort in their stomach, intestines, or bowel movements. These disorders degrade a person’s quality of life and result in significant increases in health care costs [2]. The gastrointestinal tract is an essential system for digesting food, absorbing nutrients, and excreting waste products. In this action, the movement of content in sequence is key, and this movement is supported by active and passive peristalsis (coordinated slow waves of muscle contraction and relaxation) [3,4].

*Smilacis Glabrae Rhixoma* (SGR) is an herbaceous vine widely distributed in Burma, India, Vietnam, Thailand, and China [5]. SGR is described in the Chinese Pharmacopoeia regarding the treatment of syphilis, furunculosis, dermatitis, eczema, cystitis, nephritis, and mercury poisoning [6]. Modern scientific research has demonstrated that SGR possesses several pharmacological and biological effects, including anti-inflammatory [7], antioxidant [8], antipsoriatic [9], anticancer [10,11], and antihyperuricemic [12] effects. It is known to the public that SGR is effective in helping digestion, but the specific mechanism is not yet known.

For the scientific analysis of SGR regarding gastrointestinal motility disorder, we took a network pharmacological approach integrating bioinformatics and systems biology. Traditional drugs such as SGR induce therapeutic effects based on the complexity of their ingredients and their effects on various biological systems, so the Western pharmacological approach of the existing ‘single target single drug’ is difficult to explain. However, network pharmacology is specialized in the analysis of ‘multi-target, multi-component’, suggesting a new paradigm for the scientific analysis of traditional drugs [13]. We intend to predict the components and mechanisms that SGR can treat for GMD through network pharmacological analysis of SGR, and to prove the therapeutic effect of SGR on GMD through mouse experiments. Network pharmacology analysis was conducted according to the scheme in Figure 1.

## 2. Results

### 2.1. Information of 390 Targets Derived through Correlation Investigation between Compounds and Targets

We identified 74 potential active compounds of SGR using the traditional Chinese medicine systems pharmacology database and analysis platform (TCMSP) database (Appendix A). Among them, 56 compounds had target information (Appendix A), and it was found that 56 compounds and 390 targets interact with each other by the combination of 904 compounds (Figure 2). As shown in Figure 2, quercetin was linked to the most targets (154 genes), followed by resveratrol (151 genes), succinic acid (60 genes), oleic acid (48 genes), beta-sitosterol (38 genes), naringenin (37 genes), (-)-epicatechin (32 genes), Stigmasterol (31 genes), (L)-alpha-Terpineol (22 genes), HMF (20 genes), and (R)-linalool (20 genes).

### 2.2. Seventeen Active Compounds Achieved the Criteria for Absorption, Distribution, Metabolism, and Excretion Parameters Applied

Seventeen compounds were included in the active compound screening criteria (Table 1), including (-)-taxifolin, (2R,3R)-2-(3,5-dihydroxyphenyl)-3,5,7-trihydroxychroman-4-one, 2H-3,9a-Methano-1-benzoxepin-9-methanol, octahydro-2,2,5a-trimethyl-, (3R-(3alpha,5aalpha,9alpha,9aalpha))-, 4,7-Dihydroxy-5-methoxyl-6-methyl-8-formyl-flavan, beta-sitosterol, cis-Dihydroquercetin, Dihydroresveratrol, diosgenin, EIC, Enhydrin, Methyllinolenate, naringenin, oleic acid, quercetin, sitosterol, Stigmasterol, and taxifolin.

### 2.3. Thirty-Nine Compounds Related to Gastrointestinal (GI) Disease Were Identified in SGR

We also investigated the compound–target–disease relationship using the TCMSP database. We noted that 39 compounds are associated with GI disease (Table 2). In particular, (-)-taxifolin, (2R,3R)-2-(3,5-dihydroxyphenyl)-3,5,7-trihydroxychroman-4-one, 4,7-Dihydroxy-5-methoxyl-6-methyl-8-formyl-flavan, beta-sitosterol, cis-Dihydroquercetin, Dihydroresveratrol, diosgenin, EIC, Methyllinolenate, naringenin, oleic acid, quercetin, Stigmasterol, and taxifolin were found to be active compounds associated with gastrointestinal disease, and other compounds, (-)-epicatechin, (2S,3R)-3,5,7-trihydroxy-2-(4-hydroxyphenyl)chroman-4-one, (L)-alpha-Terpineol, (R)-linalool, 3-O-caffeoylshikimic acid, Aromadedrin, astilbin, delta-amorphene, engeletin, FER, HMF, isoengelitin, Istidina, L-Bornyl acetate, myristic acid, n-butyl-β-D-fructopyronoside, n-butyl-β-D-fructoufranoside, NCA, nonane, palmitic acid, resveratrol, Sitogluside, SKM, stearic acid, and succinic acid, associated with gastrointestinal diseases, were classified as not active compounds (Figure 3).

### 2.4. Seventeen GI-Disease-Related Compounds in SGR Were Associated with GMD

To confirm the network relationship between SGR and GMD, genetic information related to GMD was checked using the cytoscape app. GMD-associated genes were found with a cutoff of 0.40 with a confidence (score) of 0.40 by limiting to a maximum of 100 proteins (Appendix A). Based on the derived data, a network of GMD-related genes and SGR target genes was created (Figure 4). There are 19 genes in both gene sets, and among the GMD-related genes, the genes targeted by SGR are ACHE, CCK, CXCL8, DPP4, FOS, GCG, IL10, IL1B, IL6, INS, MPO, OPRM1, PYY, SCN5A, SLC6A4, TNF, NOS1, HTR3A and TRPV1.

### 2.5. The Network of GMD-Related Genes and Compounds for Identification of Interesting Molecules

Figure 5 shows a network of relationships between SGR compounds and GMD-related target genes. As a result of the network analysis, it was found that quercetin, resveratrol, and SCN5A were the most related to GMD. In short, (-)-epicatechin, (L)-alpha-Terpineol, (R)-linalool, 4,7-Dihydroxy-5-methoxyl-6-methyl-8-formyl-flavan, EIC, HMF, Sitogluside, Stigmasterol, astilbin, beta-sitosterol, n-butyl-β-D-fructopyronoside, nonane, oleic acid, palmitic acid, quercetin, resveratrol, and succinic acid are active compounds targeting GMD-related genes, which means these compounds can be potential medicinal candidates.

### 2.6. Effects of SGR Extract on Pacemaker Activity on Interstitial Cells of Cajal (ICC) in Murine Small Intestine

The SGR extract depolarized pacemaker potentials and decreased the frequency (Figure 6A–C). The degrees of depolarization were 6.2 ± 0.9 mV (*p* < 0.0001) at 1 mg/mL, 10.8 ± 1.3 mV (*p* < 0.0001) at 3 mg/mL, and 25.4 ± 1.7 mV (*p* < 0.0001) at 5 mg/mL (Figure 6D). The degrees of frequency were 10.9 ± 0.8 mV at 1 mg/mL, 7.1 ± 0.8 mV (*p* < 0.0001) at 3 mg/mL, and 1.3 ± 0.4 mV (*p* < 0.0001) at 5 mg/mL (Figure 6E). These results indicated that SGR regulates the pacemaker potential of ICC.

### 2.7. Effects of SGR Extract on Delayed Gastric Emptying

Macroscopic observations showed that loperamide injecting reduced the movement of foods in the stomach, while pretreatment with the SGR extract decreased these phenomena (Figure 7B). The gastric weight of the SGR-extract-treated group was reduced compared to that of the loperamide group (* *p* < 0.05, Figure 7C). In addition, the amount of phenol red retention in the SGR-extract-pretreated group was reduced compared to that of the loperamide group (* *p* < 0.05, ** *p* < 0.01, Figure 7D). The Mosapride-pretreated group had similar results to the SGR-extract-pretreated group.

### 2.8. Effects of SGR Extract on Intestinal Transit Rate (ITR) in Normal Mice

The ITR was 46.9 ± 1.5% in normal mice (Figure 8A). The ITR values of SGR mice were 47.9 ± 2.8% at 0.01 g/kg, 51.9 ± 2.6% (*p* < 0.05), and 60.8 ± 1.5% (*p* < 0.0001) (Figure 8A).

### 2.9. Effects of SGR Extract on ITR in Mice Regarding Gastrointestinal Motility Dysfunction

We used GMD models (acetic acid (AA) models). The AA model decreased ITR (34.9 ± 3.2% vs. 47.9 ± 1.9% in normal Controls; *p* < 0.0001) (Figure 8B). However, the SGR extract at 0.01, 0.1, and 1 g/kg restored this response to 50.8 ± 1.5% (*p* < 0.0001), 53.1 ± 1.9% (*p* < 0.0001), and 56.5 ± 2.8% (*p* < 0.0001), respectively (Figure 8B). These results suggest that SGR restores the ITR in GMD mice.

## 3. Discussion

As a result of the analysis through network pharmacology based on these previous studies, 74 compounds including 17 active compounds were identified in SGR (Appendix A). Of the 74 compounds found, 56 compounds had target information, and a total of 390 target genes were collected (Appendix A). Seventeen of these compounds were associated with GMD (Figure 5), and GMD-related genes belonging to the target genes of the active compounds of SGR were ACHE, CCK, CXCL8, DPP4, FOS, GCG, IL10, IL1B, IL6, INS, MPO, OPRM1, PYY, SCN5A, SLC6A4, TNF, NOS1, HTR3A and TRPV1 (Figure 4). These results are consistent with previous studies. Specifically, SCN5A was the target of many compounds, including quercetin (Table 2, Figure 5). Nav1.5 encoded by the SCN5A gene is mediated by the rapid introduction of Na+ ions (I_Na_). Voltage-gated Na+ channels are important for excitation and propagation of electrical impulses in excitable cells, for example nerve or cardiomyocytes [14]. SCN5A is known to contribute to improving gastrointestinal motility [15]. In addition, TNF, a target gene of several compounds including quercetin, is also known to be closely related to intestinal motility [16]. In addition, FOS, a target gene for several compounds including quercetin, is considered a metabolic marker of nerve activation and is also related to gastrointestinal motility [17]. As shown in Figure 5, quercetin (11), resveratrol (8), oleic acid (5), and beta-sitosterol (4) were identified as compounds with many target genes among compounds related to GMD. Several studies have reported the relationship between key compounds and GMD. It is known that quercetin, which has the most target genes, blocks Ca2+ channels and inhibits intestinal contraction [18]. In addition, resveratrol was reported to have a function of inducing the relaxation of gastric smooth muscle through high-conductivity calcium-activated potassium channels [19]. Oleic acid is reported to slow gastrointestinal transit and reduce diarrhea by activating a nutrient-induced inhibitory feedback mechanism [20]. In this mouse study, we studied the effect of the SGR extract on pacemaker activity in ICC and gastric emptying. In addition, we examined GI function by the ITR in normal or experimentally induced GMD mice. The SGR extract depolarized the pacemaker activity in ICC (Figure 6) and restored the delayed gastric emptying (Figure 7). In addition, oral administration of the SGR extract increased the ITR and restored ITR delay (Figure 8). AA induction showed a significant decrease in the ITR. This reduction was largely blocked by the SGR extract. As a result of the network analysis of this study, SGR is strongly related to SCN5A, TNF, and FOS, which are genes highly related to intestinal motility [14,15,16,17], and oleic acid, quercetin, and resveratrol, which are major compounds of SGR also known to have an effect on intestinal motility [18,19,20]. In general, traditional herbal medicines are known to be contaminated with heavy metals, microorganisms, pesticides, etc., and can cause serious side effects [21]. In addition, if herbal medicines change in a similar shape, serious complications occur [21]. Side effects caused by SGR are not well known and are known to cause hepatotoxicity, a common side effect of herbal medicines [22]. In addition, the Chinese classic book shows that the LD50 in mice is 161 g/kg and is 100 g/kg in rats; 45 consecutive days of administration at these concentrations resulted in significant experimental animal death (*p* < 0.01), and rats’ BUN exceeded the normal range [23]. In addition, when administered at 50 g/kg to rats for 60 days, abnormalities in BUN occurred partially, but recovered after discontinuation of administration, and it was reported that degenerative necrosis appeared in hepatocytes and renal cells [23]. Therefore, in this paper, when administered at a maximum SGR of 5 g/kg for 14 days, there was no special change in important organs, so it is considered that there was no toxic reaction.

ICC are cells that control the pacemaker of the gastrointestinal tract and play an important role in regulating gastrointestinal motility [24]. Therefore, it is known that problems with ICC cause various gastrointestinal motility diseases, and ICC are an important research tool in the study of functional gastrointestinal motility regulation. In this study, it is thought that the SGR extract causes depolarization of ICC (Figure 6), and as a result of designing an experiment and confirming the gastric emptying and ITR, it was confirmed that decreased gastric emptying and intestinal mobility was restored through the SRG extract (Figure 7 and Figure 8).

It takes about 60 to 100 min for solid food and about 10 to 45 min for food containing liquid to move to the duodenum from the stomach. After about 180 min, about 70% of the food must be digested. An abnormal gastric emptying is called Gastroparesis. Gastroparesis is a syndrome that is expressed as symptoms such as nausea, vomiting, and abdominal pain due to various causes [25]. In the United States, it is observed in about 3% of the total population, and the prevalence is known to be somewhat higher in women [26]. According to recent research results, there are also reports that the prevalence rate has increased compared to the past [27]. It is reported that 30–50% of patients with type 1 diabetes and 15–30% of patients with type 2 diabetes have Gastroparesis [28], and it is on the rise recently. Intestinal motility is movement caused by the digestion of the intestines. It refers to sending food digested in the stomach to the spring intestines and large intestine. Impairment of bowel motility occurs everywhere in the gastrointestinal tract. In particular, irritable bowel syndrome (IBS) is a general abnormal condition of the intestine that shows symptoms such as abdominal pain, abdominal cramps, abdominal distension, bowel pain, and nausea [29]. The etiology of IBS is largely unknown or incomprehensible, and therefore there is no complete cure [29]. Despite the importance of gastrointestinal motility, it is difficult to choose suitable treatment because the pathophysiological stage varies due to delayed gastric emptying, hypersensitivity to gastric expansion, adverse reactions to duodenal fat or acid, and psychoneurotic disorders [30]. According to this trend, various traditional medical treatment methods and approaches to treatment of gastrointestinal motility abnormalities are being made [31]. Research on gastrointestinal motility abnormalities in traditional medicine has been actively conducted [31]. It was found that the possibility of controlling gastrointestinal movement and reduced gastrointestinal movement increased due to various traditional medicines, such as Pyungwi-san [32], Lizhong Tang [33], *Poncirus trifoliate* fruit [34], So-Cheong-Ryong-Tang [35], San-Huang-Xie-Xin-tang [36], *Schisandra chinensis* [37], Ge-Gen-Tang [38], *Liriope Platyphylla Wang Et tang* [39], Hwangryunhaedok-tang [40], Gamisoyo-San [41], Banhasasim-Tang [42], and *Pinellia ternata Breitenbach* [43]. In order for the oriental medical approach to become an efficient approach in controlling gastrointestinal motility, large-scale clinical studies are needed to prove the reproducibility and validity of the effects.

## 4. Materials and Methods

### 4.1. Network Pharmacology Analysis by SGR

#### 4.1.1. Classification of the Compound of SGR

In order to classify the potential active compounds of SGR, TCMSP databases and analysis platforms were used to investigate. We entered ‘*Smilacis Glabrae Rhixoma*’ as the query to search with the herb name.

#### 4.1.2. Analysis of Target

Target information of compounds was found using search through TCMSP [44]. We used the UniProtKB database to link the target protein to the official genetic name (https://www.uniprot.org/uniprot; accessed on 4 July 2022) [45].

#### 4.1.3. Analysis of Network

Cytoscape 3.9.2 (https://cytoscape.org; accessed on 4 July 2022) was used to build the compound–target network [46]. GMD-related genes were collected by Cytoscape App. using ‘Gastrointestinal mobility disorder’ as a query, which integrates evidence and updates weekly [47].

#### 4.1.4. Active Compound Screening

On the basis of absorption, distribution, metabolism, and excretion (ADME) parameters such as molecular weight (MW), oral bioavailability (OB), Caco-2 permeability (Caco-2), and drug likeness (DL), physiologically active compounds of SGR were screened using the following criteria: OB ≥ 30%, DL ≥ 0.10, and Caco-2 ≥ −0.4. The compounds that meet the applied criteria were recognized as active compounds.

### 4.2. Animal Validation

#### 4.2.1. SGR

SGR was purchased from Korea Plant Extract Bank (Cheongju, Republic of Korea). SGR was dissolved in distilled water and mice were administered doses in the quantities of 0.01, 0.1, and 1 g/kg. To check the toxicity of SGR extract in mice, it was administered intragastrically through an orogastric tube at different doses (0, 0.5, 1, 2, or 5 g/kg of SGR extract delivered at 10 mL/kg). Seven mice of each gender were tested, and thus, a total of 70 mice were used. Each group was carefully observed for overt clinical signs and mortality at hourly intervals for 6 h after administration, and then on a daily basis for 14 days. Individual body weights were measured before dosing and on days 1, 3, 7, and 14 after administration. At 14 days, the last day of observation, animals were necropsied, and vital organs and tissues appeared normal by gross inspection. Accordingly, the SGR extract appeared to be safe and did not elicit acute toxicity for a single oral dose of at least 5 g/kg or below. In this paper, SGR 0.01 g/kg, 0.1 g/kg, and 1 g/kg were used in the animal experiment, and this concentration has been widely used in the mouse gastrointestinal animal experiments [33,37,40,41,42,48,49,50,51]. When converted to human concentration, 1 g/kg for a mouse is 4.9 g for adults (60 kg) [52], which is less than the typical human dose of 15–60 g [23]. In addition, in rats, SGR was used at a similar concentration to this paper [53].

#### 4.2.2. Animal Experiments

A total of 48 mice (Control (*n* = 12, male 6: female 6), SGR 1 mg (*n* = 12, 6:6), SGR 3 mg (*n* = 12, 6:6), and SGR 5 mg (*n* = 12, 6:6); 4–8 days old; weighing 2.0–2.3 g) of the Institute of Cancer Research (ICR) mice from Samtako Bio Korea Co., Ltd. (Osan, Republic of Korea) were used for the ICC experiments. A total of 86 male mice (Naïve (*n* = 15), Control (*n* = 15), SGR 0.01 mg (*n* = 14), SGR 0.1 mg (*n* = 15), SGR 1 mg (*n* = 14), Mosapride (*n* = 13)); 7 weeks old; weighing 23–26 g) for the gastric emptying experiments. Also, 59 male mice (Naïve (*n* = 15), SGR 0.01 mg (*n* = 15), SGR 0.1 mg (*n* = 14), SGR 1 mg (*n* = 15)); 7 weeks old; weighing 23–26 g) for the normal ITR experiments and 74 male mice (Naïve (*n* = 15), Control (*n* = 15); SGR 0.01 mg (*n* = 15), SGR 0.1 mg (*n* = 15), SGR 1 mg (*n* = 14); 7 weeks old; weighing 23–26 g) were used for the ITR experiments with GMD. All mice were housed in a specific pathogen-free laboratory environment under controlled temperature (20 ± 2 °C) and humidity (49 ± 5%) with day and night cycles (light on at 7:00 a.m. and light off at 7:00 p.m.) and ad libitum access to normal diet and autoclaved water. During the study, indicators of the general condition of the mice were observed daily, such as fur brightness, food and water intake, defecation, and behavior. In the ICC experiments, ICC were placed under a microscope after making the cells, and SGR was administered to the cells to confirm the change in pacemaker potentials. In the gastric emptying and ITR experiments, SGR was administered directly to the mouse mouth.

#### 4.2.3. Preparation of ICC and Electrophysiological Experiments

After taking out the small intestine, the contents of the small intestine were removed with a Kreb Ringer solution. The tissues were pinned to the sylgard dish, and the mucous membrane was removed with scissors. The small intestine tissue was finely cut and then immersed in Hank solution (KCl 5.36 mM, NaCl 125 mM, NaOH 0.34 mM, Na_2_HCO_3_ 0.44 mM, Glucose 10 mM, Sucrose 2.9 mM, and Hepes 11 mM) for 30 min. The cells were then separated in a solution containing various enzymes. The cells were sprayed on collagen-coded cover glass and maintained in a 37 °C environment in media containing smooth muscle growth medium (Clonetics, San Diego, CA, USA) and stem cell factor (Sigma-Aldrich, St. Louis, MO, USA). The whole-cell patch clamp method was used to measure the change in the membrane potential in the cultured ICC. Axopatch 200B (Axon Instruments, Foster, CA, USA) was used as an amplifier, and the results were summarized using pCLAMP and Origin (version 2018; MicroCal, Northampton, MA, USA). All experiments were conducted between 30 °C and 33 °C.

#### 4.2.4. Gastric Weight and Gastric Emptying by Phenol Red

Gastric emptying was assessed by administering a 0.05% (*w*/*v*) phenol red solution (0.5 mL/mouse) 30 min after administering SGR. The mice were euthanized 30 min after treatment with phenol red, their stomachs were removed immediately, and the weights were measured. Next, the stomach was treated with 5 mL of 0.1 N sodium hydroxide solution to check the optical density of the phenol red remaining in the stomach. The homogenate was centrifuged at 3000 rpm for 20 min. Then, 1 mL of the supernatant was added to 4 mL of 0.5N sodium hydroxide solution, and the optical density of this pink liquid was measured using a spectrophotometer (560 nm). For experiments, mice were fasted for 19 h with a free supply of tap water. The selection of the phenol red solution volume (500 µL) and the 50% delayed gastric emptying time point (30 min after intraperitoneal injection of 10 mg/kg of loperamide) was conducted according to earlier study protocols [54,55]. The above emission values were obtained according to the following formula: Gastric emptying (%) = (1 − X/Y) × 100. X: Optical density of the phenol red remaining on it. Y: Optical density of the phenol red mixture with sodium hydroxide under test tube conditions.

#### 4.2.5. Intestinal Transit Rate Evaluation

Evans Blue was orally administered after SGR administration, and animals were sacrificed 30 min later. The ITR was measured by checking the length that Evans Blue passed in the intestine. The ITR was measured as a percentage of the total length passed by Evans Blue. A peritoneal irritation by the acetic acid (PIA) model, one of the experimental GMD models, was used. Peritoneal irritation was induced using acetic acid (AA) in mice 30 min after the intragastric administration of SGR. PIA was induced by an intraperitoneal (i.p.) injection of acetic acid (0.6%, *w*/*v*, in saline) at a dose of 10 mL/kg, as previously described [37,39]. After injecting acetic acid, mice were placed in individual cages and allowed to recover for 30 min.

#### 4.2.6. Statistical Analysis

The results were represented as mean ± standard error of the mean. We first checked the normality and homogeneity and then analyzed the results with ANOVA with Bonferroni post hoc tests for multiple comparisons and Prism 6.0 (La Jolla, CA, USA) programs; *p* < 0.05 was considered statistically important.

## 5. Conclusions

Alternative medicine is still not recognized as general medicine. Nevertheless, around 51% of people with gastrointestinal disorders are receiving help from alternative medicine, and 10% of all alternative medicines are for digestive problems. In addition, herbal medicines are a source of alternative medicines and are known to have fewer side effects. An association of SGR with gastrointestinal motility was found through network pharmacological analysis, and recovery of gastric emptying and intestinal motility were confirmed through mouse experiments. Through this, the great potential of SGR as a powerful traditional medicine treatment for GMD was discovered.

## Figures and Tables

**Figure 1 plants-12-01509-f001:**
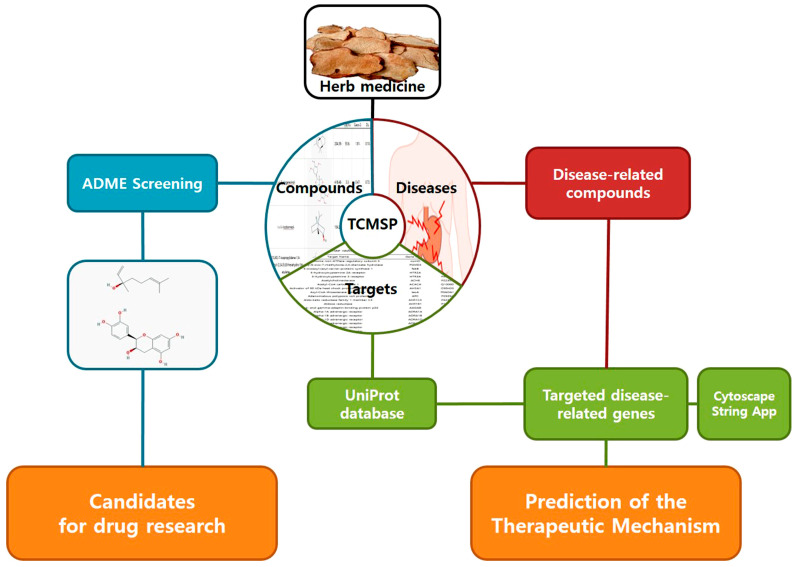
Schematic of study.

**Figure 2 plants-12-01509-f002:**
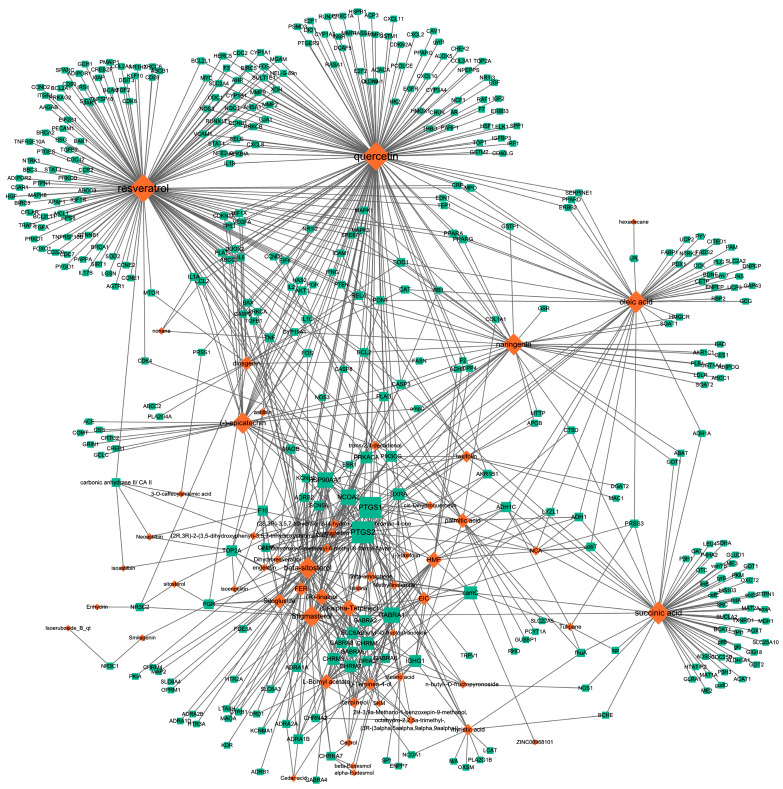
Compound–target network of SGR. The node size depends on the connected edges’ number. The compounds are expressed as orange diamond-shaped nodes, and the targets are expressed as green square-shaped nodes. SGR: *Smilacis Glabrae Rhixoma*.

**Figure 3 plants-12-01509-f003:**
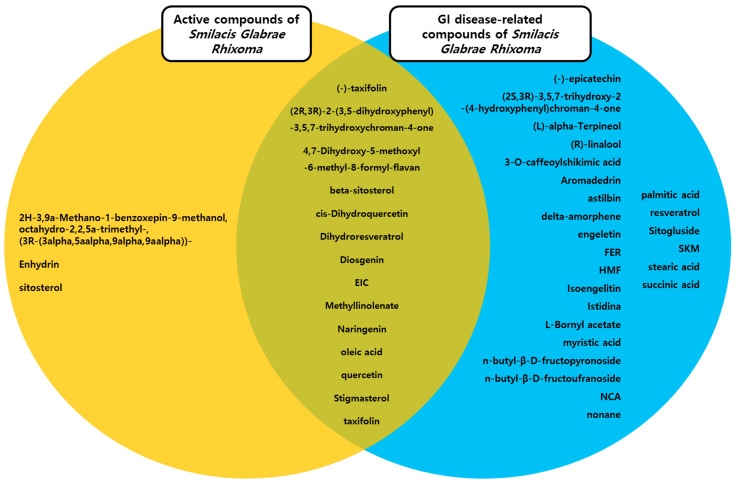
Active compounds and gastrointestinal (GI)-related compounds of SGR. The Venn diagram between active compounds and GI-disease-related compounds in SGR. SGR: *Smilacis Glabrae Rhixoma*.

**Figure 4 plants-12-01509-f004:**
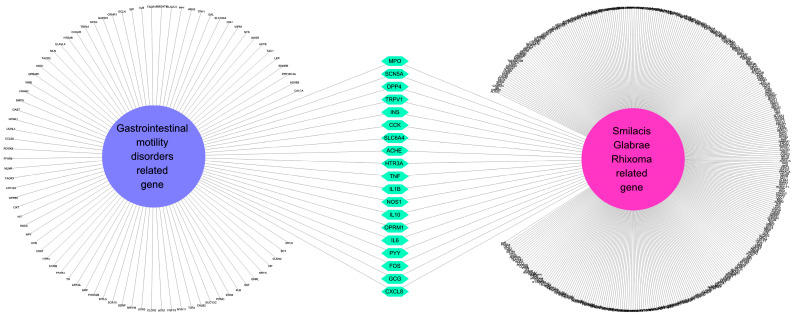
Network of GMD-related genes and SGR target genes. The 19 genes included in both “gene related to gastrointestinal motility disorder” and “*Smilacis Glabrae Rhixoma* target genes” are classified in the center. GMD: gastrointestinal motility disorder. SGR: *Smilacis Glabrae Rhixoma*.

**Figure 5 plants-12-01509-f005:**
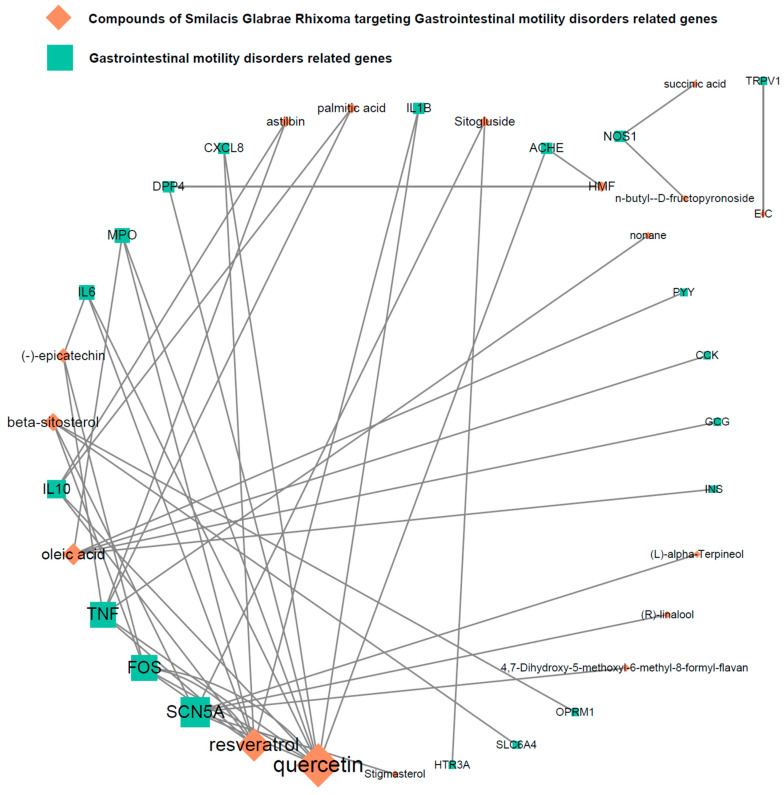
Network of compounds of SGR and GMD related genes. GMD: Gastrointestinal Motility Disorder. SGR: *Smilacis Glabrae Rhixoma*.

**Figure 6 plants-12-01509-f006:**
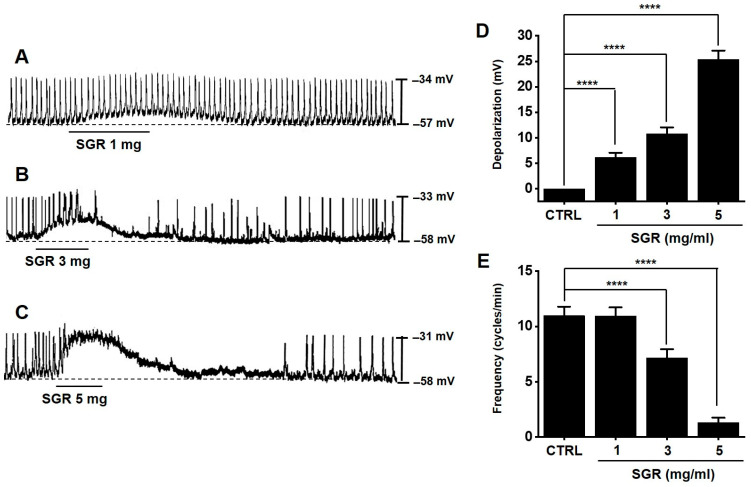
Effects of SGR extract on pacemaker activity of ICC. (**A**–**C**) SGR depolarized the ICC pacemaker activity; (**D**) the depolarization changes; (**E**) the frequency changes. CTRL (*n* = 12). SGR 1 mg (*n* = 12). SGR 3 mg (*n* = 12). SGR 5 mg (*n* = 12). Bars represent mean ± standard error. **** *p* < 0.0001. CTRL: Control. SGR: *Smilacis Glabrae Rhixoma*. ICC: interstitial cells of Cajal.

**Figure 7 plants-12-01509-f007:**
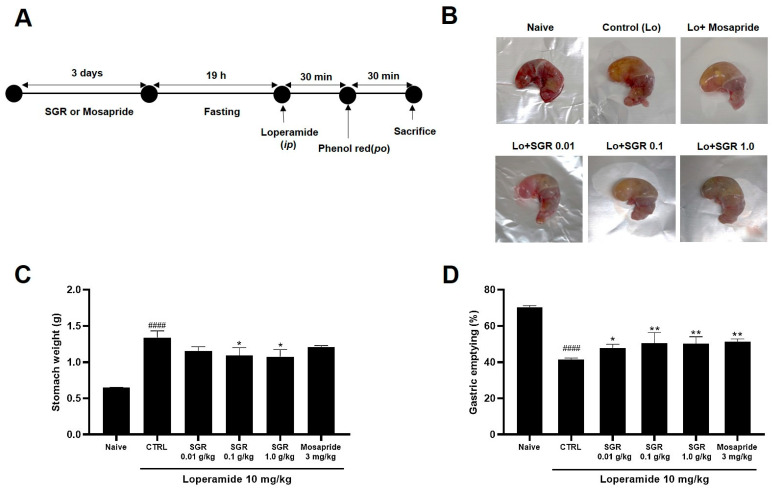
Effects of SGR extract on delayed gastric emptying. (**A**) summary of the experimental schedule; (**B**) visualization; (**C**) stomach weight; (**D**) gastric emptying. Naïve (*n* = 15). CTRL (*n* = 15). SGR 0.01 g (*n* = 14). SGR 0.1 g (*n* = 15). SGR 1 g (*n* = 14). Mosapride (*n* = 13). Bars represent mean ± standard error. * *p* < 0.05, ** *p* < 0.01 for the Control group; #### *p* < 0.0001 for the loperamide group. CTRL: Control. SGR: *Smilacis Glabrae Rhixoma*.

**Figure 8 plants-12-01509-f008:**
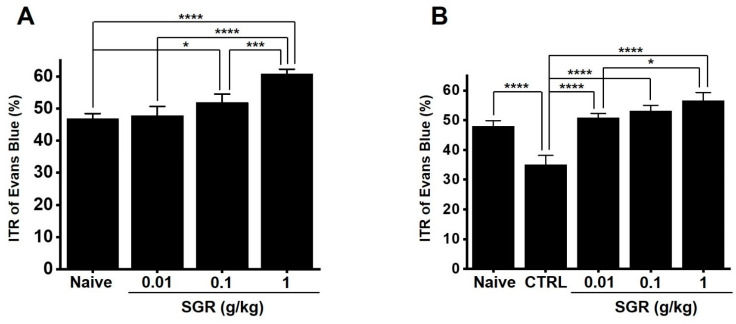
Effect of SGR extract on ITR in normal and AA-induced GMD mice. (**A**) In normal mice, SGR extract increased ITR. Naïve (*n* = 15). SGR 0.01 g (*n* = 15). SGR 0.1 g (*n* = 14). SGR 1 g (*n* = 15). (**B**) In this AA-induced case, the ITR was recovered by SGR extract. Naïve (*n* = 15). CTRL (*n* = 15). SGR 0.01 g (*n* = 15). SGR 0.1 g (*n* = 15). SGR 1 g (*n* = 14). Bars represent mean ± standard error. * *p* < 0.05. *** *p* < 0.001. **** *p* < 0.0001. CTRL: Control. SGR: *Smilacis Glabrae Rhixoma*. ITR: intestinal transit rate. AA: acetic acid. GMD: gastrointestinal motility disorder.

**Table 1 plants-12-01509-t001:** Active compounds of *Smilacis Glabrae Rhixoma*.

Molecule Name	Structure	MW *	OB (%) *	Caco-2 *	DL *
(-)-taxifolin	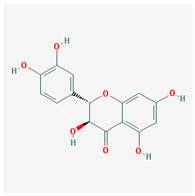	304.27	60.51	−0.24	0.27
(2R,3R)-2-(3,5-dihydroxyphenyl)-3,5,7-trihydroxychroman-4-one	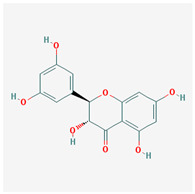	304.27	63.17	−0.34	0.27
2H-3,9a-Methano-1-benzoxepin-9-methanol, octahydro-2,2,5a-trimethyl-, (3R-(3alpha,5aalpha,9alpha,9aalpha))-	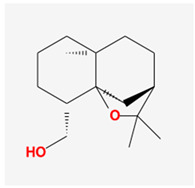	238.41	98.38	1.01	0.14
4,7-Dihydroxy-5-methoxyl-6-methyl-8-formyl-flavan	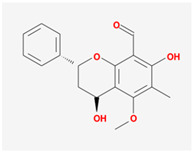	314.36	37.03	0.48	0.28
beta-sitosterol	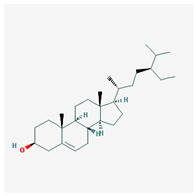	414.79	36.91	1.32	0.75
cis-Dihydroquercetin	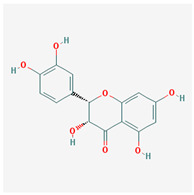	304.27	66.44	−0.34	0.27
Dihydroresveratrol	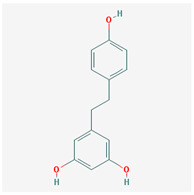	230.28	87.27	0.81	0.11
diosgenin	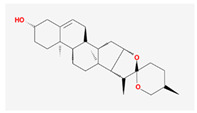	414.69	80.88	0.82	0.81
EIC	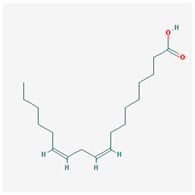	280.5	41.9	1.16	0.14
Enhydrin	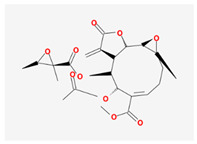	464.51	40.56	−0.36	0.74
Methyllinolenate	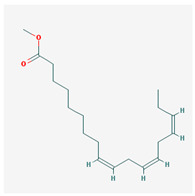	292.51	46.15	1.48	0.17
naringenin	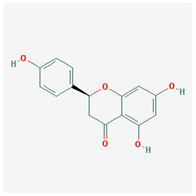	272.27	59.29	0.28	0.21
oleic acid	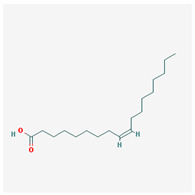	282.52	33.13	1.17	0.14
quercetin	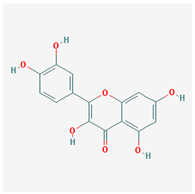	302.25	46.43	0.05	0.28
sitosterol	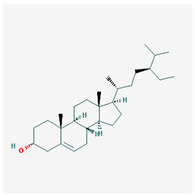	414.79	36.91	1.32	0.75
Stigmasterol	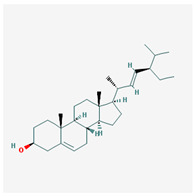	412.77	43.83	1.44	0.76
taxifolin	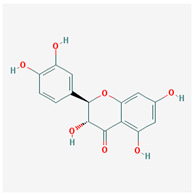	304.27	57.84	−0.23	0.27

* Molecular weight (MW), oral bioavailability (OB), Caco-2 permeability (Caco-2), and drug likeness (DL).

**Table 2 plants-12-01509-t002:** Compounds and targets related to gastrointestinal disease.

Molecule Name	Target Name	Disease Name
(-)-epicatechin	Heat shock protein HSP 90	Gastrointestinal Stromal Tumors (GIST)
	Interleukin-6	* Gastrointestinal motility disorders
	Prostaglandin G/H synthase 2	Adenomatous polyposis
		Colorectal cancer
		Oropharyngeal squamous cell carcinoma
		Peutz–Jeghers syndrome
	Transcription factor AP-1	* Gastrointestinal motility disorders
	Tumor necrosis factor	* Gastrointestinal motility disorders
		Crohns’s Disease, unspecified
(-)-taxifolin	Heat shock protein HSP 90	Gastrointestinal Stromal Tumors (GIST)
	Prostaglandin G/H synthase 2	Adenomatous polyposis
		Colorectal cancer
		Oropharyngeal squamous cell carcinoma
		Peutz–Jeghers syndrome
(2R,3R)-2-(3,5-dihydroxyphenyl)-3,5,7-trihydroxychroman-4-one	Heat shock protein HSP 90	Gastrointestinal Stromal Tumors (GIST)
	Prostaglandin G/H synthase 2	Adenomatous polyposis
		Colorectal cancer
		Oropharyngeal squamous cell carcinoma
		Peutz–Jeghers syndrome
(2S,3R)-3,5,7-trihydroxy-2-(4-hydroxyphenyl)chroman-4-one	Heat shock protein HSP 90	Gastrointestinal Stromal Tumors (GIST)
	Prostaglandin G/H synthase 2	Adenomatous polyposis
		Colorectal cancer
		Oropharyngeal squamous cell carcinoma
		Peutz–Jeghers syndrome
(L)-alpha-Terpineol	Heat shock protein HSP 90	Gastrointestinal Stromal Tumors (GIST)
	Prostaglandin G/H synthase 2	Adenomatous polyposis
		Colorectal cancer
		Oropharyngeal squamous cell carcinoma
		Peutz–Jeghers syndrome
	Sodium channel protein type 5 subunit alpha	* Gastrointestinal motility disorders
(R)-linalool	Heat shock protein HSP 90	Gastrointestinal Stromal Tumors (GIST)
	Prostaglandin G/H synthase 2	Adenomatous polyposis
		Colorectal cancer
		Oropharyngeal squamous cell carcinoma
		Peutz–Jeghers syndrome
	Sodium channel protein type 5 subunit alpha	* Gastrointestinal motility disorders
3-O-caffeoylshikimic acid	Prostaglandin G/H synthase 2	Adenomatous polyposis
		Colorectal cancer
		Oropharyngeal squamous cell carcinoma
		Peutz–Jeghers syndrome
4,7-Dihydroxy-5-methoxyl-6-methyl-8-formyl-flavan	Heat shock protein HSP 90	Gastrointestinal Stromal Tumors (GIST)
	Prostaglandin G/H synthase 2	Adenomatous polyposis
		Colorectal cancer
		Oropharyngeal squamous cell carcinoma
		Peutz-Jeghers syndrome
	Sodium channel protein type 5 subunit alpha	* Gastrointestinal motility disorders
Aromadedrin	Heat shock protein HSP 90	Gastrointestinal Stromal Tumors (GIST)
	Prostaglandin G/H synthase 2	Adenomatous polyposis
		Colorectal cancer
		Oropharyngeal squamous cell carcinoma
		Peutz–Jeghers syndrome
astilbin	Interleukin-10	* Gastrointestinal motility disorders
	Tumor necrosis factor	* Gastrointestinal motility disorders
		Crohns’s Disease, unspecified
beta-sitosterol	Heat shock protein HSP 90	Gastrointestinal Stromal Tumors (GIST)
	Mu-type opioid receptor	* Gastrointestinal motility disorders
		Diarrhea
		Opioid-induced bowel dysfunction
	Prostaglandin G/H synthase 2	Adenomatous polyposis
		Colorectal cancer
		Oropharyngeal squamous cell carcinoma
		Peutz-Jeghers syndrome
	Sodium channel protein type 5 subunit alpha	* Gastrointestinal motility disorders
	Sodium-dependent serotonin transporter	* Gastrointestinal motility disorders
	Transcription factor AP-1	* Gastrointestinal motility disorders
cis-Dihydroquercetin	Heat shock protein HSP 90	Gastrointestinal Stromal Tumors (GIST)
	Prostaglandin G/H synthase 2	Adenomatous polyposis
		Colorectal cancer
		Oropharyngeal squamous cell carcinoma
		Peutz–Jeghers syndrome
delta-amorphene	Prostaglandin G/H synthase 2	Adenomatous polyposis
		Colorectal cancer
		Oropharyngeal squamous cell carcinoma
		Peutz–Jeghers syndrome
Dihydroresveratrol	Heat shock protein HSP 90	Gastrointestinal Stromal Tumors (GIST)
	Prostaglandin G/H synthase 2	Adenomatous polyposis
		Colorectal cancer
		Oropharyngeal squamous cell carcinoma
		Peutz–Jeghers syndrome
diosgenin	Prostaglandin G/H synthase 2	Adenomatous polyposis
		Colorectal cancer
		Oropharyngeal squamous cell carcinoma
		Peutz–Jeghers syndrome
	Vascular endothelial growth factor A	Colorectal Neoplasms
EIC	Prostaglandin G/H synthase 2	Adenomatous polyposis
		Colorectal cancer
		Oropharyngeal squamous cell carcinoma
		Peutz–Jeghers syndrome
	Transient receptor potential cation channel subfamily V member 1	* Gastrointestinal motility disorders
engeletin	Heat shock protein HSP 90	Gastrointestinal Stromal Tumors (GIST)
	Prostaglandin G/H synthase 2	Adenomatous polyposis
		Colorectal cancer
		Oropharyngeal squamous cell carcinoma
		Peutz–Jeghers syndrome
FER	Heat shock protein HSP 90	Gastrointestinal Stromal Tumors (GIST)
	Leukotriene A-4 hydrolase	Esophageal cancer
	Nitric oxide synthase, endothelial	Colon cancer
	Prostaglandin G/H synthase 2	Adenomatous polyposis
		Colorectal cancer
		Oropharyngeal squamous cell carcinoma
		Peutz–Jeghers syndrome
HMF	Acetylcholinesterase	* Gastrointestinal motility disorders
	Dipeptidyl peptidase IV	* Gastrointestinal motility disorders
	Prostaglandin G/H synthase 2	Adenomatous polyposis
		Colorectal cancer
		Oropharyngeal squamous cell carcinoma
		Peutz–Jeghers syndrome
isoengelitin	Prostaglandin G/H synthase 2	Adenomatous polyposis
		Colorectal cancer
		Oropharyngeal squamous cell carcinoma
		Peutz–Jeghers syndrome
Istidina	Prostaglandin G/H synthase 2	Adenomatous polyposis
		Colorectal cancer
		Oropharyngeal squamous cell carcinoma
		Peutz–Jeghers syndrome
L-Bornyl acetate	Prostaglandin G/H synthase 2	Adenomatous polyposis
		Colorectal cancer
		Oropharyngeal squamous cell carcinoma
		Peutz–Jeghers syndrome
Methyllinolenate	Prostaglandin G/H synthase 2	Adenomatous polyposis
		Colorectal cancer
		Oropharyngeal squamous cell carcinoma
		Peutz–Jeghers syndrome
myristic acid	Prostaglandin G/H synthase 2	Adenomatous polyposis
		Colorectal cancer
		Oropharyngeal squamous cell carcinoma
		Peutz–Jeghers syndrome
naringenin	Glutathione S-transferase P	Colorectal Neoplasms
		Gastrointestinal Neoplasms
		Rectal Neoplasms
	Heat shock protein HSP 90	Gastrointestinal Stromal Tumors (GIST)
	Peroxisome proliferator-activated receptor gamma	Crohns’s Disease, unspecified
		Inflammatory Bowel Disease
		Ulcerative colitis
	Prostaglandin G/H synthase 2	Adenomatous polyposis
		Colorectal cancer
		Oropharyngeal squamous cell carcinoma
		Peutz–Jeghers syndrome
n-butyl-β-D-fructopyronoside	Nitric oxide synthase, brain	* Gastrointestinal motility disorders
	Prostaglandin G/H synthase 2	Adenomatous polyposis
		Colorectal cancer
		Oropharyngeal squamous cell carcinoma
		Peutz–Jeghers syndrome
n-butyl-β-D-fructoufranoside	Prostaglandin G/H synthase 2	Adenomatous polyposis
		Colorectal cancer
		Oropharyngeal squamous cell carcinoma
		Peutz–Jeghers syndrome
NCA	Prostaglandin G/H synthase 2	Adenomatous polyposis
		Colorectal cancer
		Oropharyngeal squamous cell carcinoma
		Peutz–Jeghers syndrome
nonane	Tumor necrosis factor	* Gastrointestinal motility disorders
		Crohns’s Disease, unspecified
oleic acid	Cholecystokinin	* Gastrointestinal motility disorders
	Glucagon	* Gastrointestinal motility disorders
	Insulin	* Gastrointestinal motility disorders
	Myeloperoxidase	* Gastrointestinal motility disorders
	Peptide YY	* Gastrointestinal motility disorders
	Peroxisome proliferator-activated receptor gamma	Crohns’s Disease, unspecified
		Inflammatory Bowel Disease
		Ulcerative colitis
	Prostaglandin G/H synthase 2	Adenomatous polyposis
		Colorectal cancer
		Oropharyngeal squamous cell carcinoma
		Peutz–Jeghers syndrome
palmitic acid	Interleukin-10	* Gastrointestinal motility disorders
	Prostaglandin G/H synthase 2	Adenomatous polyposis
		Colorectal cancer
		Oropharyngeal squamous cell carcinoma
		Peutz–Jeghers syndrome
	Tumor necrosis factor	* Gastrointestinal motility disorders
		Crohns’s Disease, unspecified
quercetin	Acetylcholinesterase	* Gastrointestinal motility disorders
	Arachidonate 5-lipoxygenase	Gastrointestinal cancers
		Inflammatory Bowel Disease
		Ulcerative colitis
	Cytochrome P450 1A2	Colorectal Neoplasms
	Dipeptidyl peptidase IV	* Gastrointestinal motility disorders
	Epidermal growth factor receptor	Colorectal cancer
	Glutathione S-transferase P	Colorectal Neoplasms
		Gastrointestinal Neoplasms
		Rectal Neoplasms
	Heat shock protein HSP 90	Gastrointestinal Stromal Tumors (GIST)
	Interleukin-1 beta	* Gastrointestinal motility disorders
	Interleukin-10	* Gastrointestinal motility disorders
	Interleukin-6	* Gastrointestinal motility disorders
	Interleukin-8	* Gastrointestinal motility disorders
	Myeloperoxidase	* Gastrointestinal motility disorders
	Nitric oxide synthase, endothelial	Colon cancer
	Ornithine decarboxylase	Hereditary Polyposis Syndromes
	Peroxisome proliferator-activated receptor gamma	Crohns’s Disease, unspecified
		Inflammatory Bowel Disease
		Ulcerative colitis
	Pro-epidermal growth factor	Colorectal Neoplasms
		Rectal Neoplasms
	Prostaglandin G/H synthase 2	Adenomatous polyposis
		Colorectal cancer
		Oropharyngeal squamous cell carcinoma
		Peutz-Jeghers syndrome
	Proto-oncogene c-Fos	* Gastrointestinal motility disorders
	Sodium channel protein type 5 subunit alpha	* Gastrointestinal motility disorders
	Thrombomodulin	Radiation enteropathy
	Transcription factor AP-1	* Gastrointestinal motility disorders
	Tumor necrosis factor	* Gastrointestinal motility disorders
		Crohns’s Disease, unspecified
	Vascular endothelial growth factor A	Colorectal Neoplasms
resveratrol	Heat shock protein HSP 90	Gastrointestinal Stromal Tumors (GIST)
	Interleukin-1 beta	* Gastrointestinal motility disorders
	Interleukin-10	* Gastrointestinal motility disorders
	Interleukin-6	* Gastrointestinal motility disorders
	Interleukin-8	* Gastrointestinal motility disorders
	Mitogen-activated protein kinase 8	Crohns’s Disease, unspecified
	Myeloperoxidase	* Gastrointestinal motility disorders
	Nitric oxide synthase, endothelial	Colon cancer
	Ornithine decarboxylase	Hereditary Polyposis Syndromes
	Peroxisome proliferator-activated receptor gamma	Crohns’s Disease, unspecified
		Inflammatory Bowel Disease
		Ulcerative colitis
	Prostaglandin G/H synthase 2	Adenomatous polyposis
		Colorectal cancer
		Oropharyngeal squamous cell carcinoma
		Peutz–Jeghers syndrome
	Proto-oncogene c-Fos	* Gastrointestinal motility disorders
	Transcription factor AP-1	* Gastrointestinal motility disorders
	Tumor necrosis factor	* Gastrointestinal motility disorders
		Crohns’s Disease, unspecified
	Vascular endothelial growth factor A	Colorectal Neoplasms
Sitogluside	5-hydroxytryptamine receptor 3A	* Gastrointestinal motility disorders
		Diarrhea
		Irritable bowel syndrome
		Postoperative nausea and vomiting
	Heat shock protein HSP 90	Gastrointestinal Stromal Tumors (GIST)
	Prostaglandin G/H synthase 2	Adenomatous polyposis
		Colorectal cancer
		Oropharyngeal squamous cell carcinoma
		Peutz–Jeghers syndrome
	Sodium channel protein type 5 subunit alpha	* Gastrointestinal motility disorders
SKM	Prostaglandin G/H synthase 2	Adenomatous polyposis
		Colorectal cancer
		Oropharyngeal squamous cell carcinoma
		Peutz–Jeghers syndrome
stearic acid	Prostaglandin G/H synthase 2	Adenomatous polyposis
		Colorectal cancer
		Oropharyngeal squamous cell carcinoma
		Peutz–Jeghers syndrome
Stigmasterol	Leukotriene A-4 hydrolase	Esophageal cancer
	Prostaglandin G/H synthase 2	Adenomatous polyposis
		Colorectal cancer
		Oropharyngeal squamous cell carcinoma
		Peutz–Jeghers syndrome
	Sodium channel protein type 5 subunit alpha	* Gastrointestinal motility disorders
succinic acid	Nitric oxide synthase, brain	* Gastrointestinal motility disorders
taxifolin	Heat shock protein HSP 90	Gastrointestinal Stromal Tumors (GIST)
	Prostaglandin G/H synthase 2	Adenomatous polyposis
		Colorectal cancer
		Oropharyngeal squamous cell carcinoma
		Peutz–Jeghers syndrome

* After investigating the relationship between *Smilacis Glabrae Rhixoma* and GI motility disorder using Cytoscape stringApp, genes related to GI motility disorder were added to this table.

## Data Availability

The datasets used and/or analyzed during the current study are available from the corresponding author upon reasonable request.

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
