# Peer review of "Network Pharmacological Analysis and Experimental Validation of the Effect of Smilacis Glabrae Rhixoma on Gastrointestinal Motility Disorder"

_plants, 2023, doi:10.3390/plants12071509_

Round 1
Reviewer 1 Report
Network pharmacological analysis and experimental validation of the effect of Smilacis Glabrae Rhixoma on gastrointestinal motility disorder
How the authors determine the active constituents of Smilacis Glabrae Rhixoma. Phytochemical analysis if this extract is highly recommended.
How did the authors determine gastrointestinal motility disorder-related genes?
Did the author target gastrointestinal motility disorder-related proteins as drug targets?
Histopathology and immunohistochemistry of the target protein should be evaluated.
Author Response
Reviewer1
Network pharmacological analysis and experimental validation of the effect of Smilacis Glabrae Rhixoma on gastrointestinal motility disorder
How the authors determine the active constituents of Smilacis Glabrae Rhixoma. Phytochemical analysis if this extract is highly recommended.
Responses) TCMSP database was used to check the compounds of SGR, and among them, compounds that satisfy oral bioavailability (OB)≥30%, drug likeness (DL)≥0.10, and Caco-2 permeability (Caco-2)≥-0.4 were set as active compounds.
How did the authors determine gastrointestinal motility disorder-related genes?
Responses) After running Cytoscape, select File-Import-Network from public database from the top bar of the program and select Data Source as STRING: PubMedquery in the pop-up window. And, enter "Gastrointestinal mobility disorder" in Pubmed Query and set Confidence (score) cutoff: 0.40, Maximum number of proteins: 100. After the import and analysis are completed, select File-Export-Table to file from the top bar of the program to save the gene list.
Did the author target gastrointestinal motility disorder-related proteins as drug targets?
Responses) We targeted genes related to gastrointestinal motility disorders as drug targets.
Histopathology and immunohistochemistry of the target protein should be evaluated.
Responses) This study is a paper to find genes related to gastrointestinal motility of SRG at the gene level and to experimentally confirm the efficacy of treating gastrointestinal motility disorders. As the author pointed out, we are currently looking for related proteins and will prove them by various experimental methods.

Reviewer 2 Report
The manuscript entitled “Network pharmacological analysis and experimental validation of the effect of Smilacis Glabrae Rhixoma on gastrointestinal motility disorder” addresses the beneficial effect of the extract of Smilacis Glabrae Rhixoma (SGR), a traditional herbal medicine, against the gastro-intestinal motility disorder (GMD) using network pharmacology analysis. Moreover, the study investigated the impact of SGR extract on the pacemaker activity in the interstitial cells of Cajal (ICC) in vitro and gastric emptying in mice in vivo. Initially, the authors demonstrated – using network pharmacology - that quercetin, resveratrol, SCN5A, TNF, and FOS were most closely related to GMD. Then, the authors proceeded to show that the SGR extract regulated the pacemaker activity in ICC and boosted delayed gastric emptying. In the same direction, SGR extract administration to acetic acid-induced GMD in mice demonstrated the competence of SGR extract to counteract the lowered intestinal transit rate in mice. The current findings are interesting.
Comments:
1) In the material and methods section or at least in the supplementary material, the authors are advised to add all the chemical charts (e.g., HPLC, NMR, ….) that confirm the identity of isolated compounds in the extract of Smilacis Glabrae Rhixoma. Without these data, the results of the study would be unreliable.
2) In line 292, the full name of the ICR abbreviation should be described at the first mention.
In the entire manuscript, please explain each abbreviation only once, when used for the first time, then use an abbreviation consequently.
3) In the material and methods section, the authors are advised to add a separate section for animal experiments. In that section, please provide the species, strain, sex, weight, and source of the animals.
4) What is the LD50 for Smilacis Glabrae Rhixoma extract in mice? Is the extract under investigation safe at the in vivo doses used? Check the literature to see if toxicological data are available. Alternatively, try to provide early safety data. If you are not able to provide such data (and such data are not available in the literature), please explain the reasons and discuss in your paper the lack of toxicity data as a limitation of the study. Please, add the answer to section 4.2.1.
5) In section 4.2.1, how were the doses of the extract (0.01, 0.1, and 1 g/kg) selected? Are the selected doses in rats relevant for human translation? Can you discuss the dose used for possible translation in humans, for example, by using conversion tables available in the literature using the Human effective dose (HED) formula= animal dose x animal Km/ human Km (Nair AB, Jacob S. A simple practice guide for dose conversion between animals and humans. J Basic Clin Pharm. 2016 Mar;7(2):27-31). Authors are advised to address this point and add the answers/proper citations to section 4.2.1.
6) In the material and methods section, please describe how many mice were in each experimental group. Please, elaborate on the description of each group and what treatments were received including, each group name (normal, control, 0.01, 0.1, and 1 g/kg), the dose received, route for administration, and duration of administration of the extract.
Please, also comment on the adequacy of the sample size calculation. How did the authors decide on using the listed number of animals per experimental group? Authors are advised to address this point and add the answers/proper citations to a separate section named as the animal experimental design.
7) To avoid readers’ confusion, the current animal study design should better be presented in a separate figure in the material and methods section.
8) Section 4.2.4. and 4.2.5. should be completely rewritten again. All the details of the animal groups, treatments, etc. must be detailed. It is not sufficient to just add the references; details of the experimental work should be described in adequate detail.
9) Since the study involves several experimental groups/treatments, statistical analysis is typically analyzed by ANOVA followed by a post-hoc test e.g., Tukey-Kramer. Please, add the name of the post-hoc test used by the authors.
10) In the statistical analysis section, did the authors check data normality and homogeneity before proceeding to one-way ANOVA? Authors are advised to address this point and add the answers to the comment in the material and methods section.
11) To make all figure legends stand-alone, authors are advised to add the full name of the used abbreviations at the end of each legend including SGR, GMD, ICC, etc. Authors are advised to address this point and add the answers to the relevant figure legends.
12) In Figures 6-8, the figure captions must include information on the number of animals/replicates from which data were extracted. Authors are advised to address this point and add the answers to the relevant figure legends.
13) In the discussion section, authors are advised to describe the reported adverse effects of the extract from the literature.
14) The manuscript needs to be carefully checked by a native English speaker for grammar and typos. Some typos/syntax errors are present in the manuscript which need to be addressed, for example:
- In line 67, the authors state that “Network pharmacology analysis was conducted according to the schematic in Figure 1”.
Please, consider correcting the above statement to “Network pharmacology analysis was conducted according to the scheme in Figure 1”.
Author Response
Reviewer2
The manuscript entitled “Network pharmacological analysis and experimental validation of the effect of Smilacis Glabrae Rhixoma on gastrointestinal motility disorder” addresses the beneficial effect of the extract of Smilacis Glabrae Rhixoma (SGR), a traditional herbal medicine, against the gastro-intestinal motility disorder (GMD) using network pharmacology analysis. Moreover, the study investigated the impact of SGR extract on the pacemaker activity in the interstitial cells of Cajal (ICC) in vitro and gastric emptying in mice in vivo. Initially, the authors demonstrated – using network pharmacology - that quercetin, resveratrol, SCN5A, TNF, and FOS were most closely related to GMD. Then, the authors proceeded to show that the SGR extract regulated the pacemaker activity in ICC and boosted delayed gastric emptying. In the same direction, SGR extract administration to acetic acid-induced GMD in mice demonstrated the competence of SGR extract to counteract the lowered intestinal transit rate in mice. The current findings are interesting.
Comments:
1) In the material and methods section or at least in the supplementary material, the authors are advised to add all the chemical charts (e.g., HPLC, NMR, ….) that confirm the identity of isolated compounds in the extract of Smilacis Glabrae Rhixoma. Without these data, the results of the study would be unreliable.
Responses) As mentioned in materials and methods, SRG compounds are confirmed by The traditional Chinese medicine systems pharmacology database and analysis platform (TCMSP; https://tcmsp-e.com/tcmsp.php). Below are recent papers published using TCMSP.
<References>
TCMSP: a database of systems pharmacology for drug discovery from herbal medicines. Jinlong Ru, Peng Li, Jinan Wang, Wei Zhou, Bohui Li, Chao Huang, Pidong Li, Zihu Guo, Weiyang Tao, Yinfeng Yang, Xue Xu, Yan Li, Yonghua Wang, Ling Yang. J Cheminform. 2014 Apr 16;6:13.
Identification of phytochemical compounds of Fagopyrum dibotrys and their targets by metabolomics, network pharmacology and molecular docking studies. Zhang M, Zhang X, Pei J, Guo B, Zhang G, Li M, Huang L. Heliyon. 2023 Mar;9(3):e14029.
Pharmacological network analysis of the functions and mechanism of kaempferol from Du Zhong in intervertebral disc degeneration (IDD). Wang X, Tan Y, Liu F, Wang J, Liu F, Zhang Q, Li J. J Orthop Translat. 2023 Mar 3;39:135-146.
Based on network pharmacology and bioinformatics to analyze the mechanism of action of Astragalus membranaceus in the treatment of vitiligo and COVID-19. Wang Y, Ding M, Chi J, Wang T, Zhang Y, Li Z, Li Q. Sci Rep. 2023 Mar 8;13(1):3884.
Integrating Network Pharmacology and Bioinformatics to Explore the Effects of Dangshen (Codonopsis pilosula) Against Hepatocellular Carcinoma: Validation Based on the Active Compound Luteolin. Yu Y, Ding S, Xu X, Yan D, Fan Y, Ruan B, Zhang X, Zheng L, Jie W, Zheng S. Drug Des Devel Ther. 2023 Mar 1;17:659-673.
Investigation on the Mechanisms of Zanthoxylum bungeanum for Treating Diabetes Mellitus Based on Network Pharmacology, Molecular Docking, and Experiment Verification. Huang Y, Gong Z, Yan C, Zheng K, Zhang L, Li J, Liang E, Zhang L, Mao J. Biomed Res Int. 2023 Feb 22;2023:9298728.
Scutellaria baicalensis in the Treatment of Hepatocellular Carcinoma: Network Pharmacology Analysis and Experimental Validation. Cai X, Peng S, Wang L, Tang D, Zhang P. Evid Based Complement Alternat Med. 2023 Feb 23;2023:4572660.
Integration of LC-LTQ-Orbitrap-MS and Network Pharmacology to Analyze the Active Components of Sijunzi Decoction and their Mechanism of Action Against Cytotoxicity-associated Premature Ovarian Insufficiency. Chen Y, Han S, Kang A, Fu R, Chen L, Guo J, Wang Q. Comb Chem High Throughput Screen. 2023 Mar 3.
2) In line 292, the full name of the ICR abbreviation should be described at the first mention.
In the entire manuscript, please explain each abbreviation only once, when used for the first time, then use an abbreviation consequently.
Responses) When I first mentioned it, I wrote it in full name. It was confirmed throughout the paper.
3) In the material and methods section, the authors are advised to add a separate section for animal experiments. In that section, please provide the species, strain, sex, weight, and source of the animals.
Responses) We added a separate section for animal experiments in the material and methods section.
A total of 48 mice (Control (n=12, male 6: female 6), SGR 1 mg (n=12, 6:6), SGR 3 mg (n=12, 6:6), SGR 5 mg (n=12, 6:6)); 4‑8 days old; weighing 2.0‑2.3 g) of the Institute of Cancer Research (ICR) mice from the Samtako Bio Korea Co., Ltd. (Osan, Republic of Korea) were used for the ICC experiments, 86 male mice (Naïve (n=15), Control (n=15), SGR 0.01 mg (n=14), SGR 0.1 mg (n=15), SGR 1 mg (n=14), Mosapride (n=13)); 7 weeks old; weighing 23‑26 g) for the gastric emptying experiments. Also, 59 male mice (Naive (n=15), SGR 0.01 mg (n=15), SGR 0.1 mg (n=14), SGR 1 mg (n=15)); 7 weeks old; weighing 23‑26 g) for the normal ITR experiments and 74 male mice (Naive (n=15), Control (n=15); SGR 0.01 mg (n=15), SGR 0.1 mg (n=15), SGR 1 mg (n=14); 7 weeks old; weighing 23‑26 g) for the ITR experiments with GMD were used. All mice were housed in a specific pathogen‑free laboratory environment under a controlled temperature (20±2ËšC) and humidity (49±5%) with day and night cycles (light on at 7:00 a.m. and light off at 7:00 p.m) and ad libitum access to normal diet and autoclaved water. During the study, indicators of the general condition of the mice were observed daily, such as fur brightness, food and water intake, defecation and behavior. In the ICC experiments, ICC were placed under a microscope after making the cells, and SGR were administered to the cells to confirm the depolarization, and in the gastric emptying and ITR experiments, SGR were administered directly to the mouse mouth.
4) What is the LD50 for Smilacis Glabrae Rhixoma extract in mice? Is the extract under investigation safe at the in vivo doses used? Check the literature to see if toxicological data are available. Alternatively, try to provide early safety data. If you are not able to provide such data (and such data are not available in the literature), please explain the reasons and discuss in your paper the lack of toxicity data as a limitation of the study. Please, add the answer to section 4.2.1.
Responses) LD50=3.2 mg in ICC experiments and mice was safe at the in vivo doses used under investigations. Toxicological experiments have been carried out during the experiment. The results were added.
To check the toxicity of SGR extract in mice, it was administered intragastrically through an orogastric tube at different doses (0, 0.5, 1, 2, or 5 g/kg of SGR extract delivered at 10 ml/kg). Seven mice of each gender were tested, and thus, a total of 70 mice were used. Each group was carefully observed for overt clinical signs and mortality at hourly intervals for 6 h after administration, and then on a daily basis for 14 days. Individual body weights were measured before dosing and on days 1, 3, 7, and 14 after administration. At 14 days, the last day of observation, animals were necropsied and vital organs and tissues appeared normal by gross inspection. Accordingly, SGR extract appear to be safe and neither elicits acute toxicity for a single oral dose of at least 5 g/kg or below.
5) In section 4.2.1, how were the doses of the extract (0.01, 0.1, and 1 g/kg) selected? Are the selected doses in rats relevant for human translation? Can you discuss the dose used for possible translation in humans, for example, by using conversion tables available in the literature using the Human effective dose (HED) formula= animal dose x animal Km/ human Km (Nair AB, Jacob S. A simple practice guide for dose conversion between animals and humans. J Basic Clin Pharm. 2016 Mar;7(2):27-31). Authors are advised to address this point and add the answers/proper citations to section 4.2.1.
Responses) In this paper, SGR 0.01 g/kg, 0.1 g/kg, and 1 g/kg were used in the animal experiment, and this concentration was widely used in the mouse gastrointestinal animal experiments [37,40-42,48-52]. When converted to human concentration, 1 g/kg of mouse is 4.9 g for adults (60 kg) [53], which is less than the typical human dose of 15-60 g [23]. In addition, in rat, SGR was used at a similar concentration to this paper [54].
<References>
23.Zhang, M.Z.; Dong, Z.H.; She, J. Modern Study Of Traditional Chinese Medicine, Vol. 1.; Xueyuan Press: Beijing, China, 1997; p.316.
37.Ahn, T.S.; Kim, D.G.; Hong, N.R.; Park, H.S.; Kim, H.; Ha, K.T.; Jeon, J.H.; So, I.; Kim, B.J. Effects of Schisandra chinensis extract on gastrointestinal motility in mice. J. Ethnopharmacol. 2015, 169, 163-169.
40.Kim, H.; Kim, I.; Lee, M.C.; Kim, H.J.; Lee, G.S.; Kim, H.; Kim, B.J. Effects of Hwangryunhaedok-tang on gastrointestinal motility function in mice. World J. Gastroenterol. 2017, 23, 2705-2715.
41.Shin, S.J.; Kim, D.; Kim, J.S.; Kim, I.; Lee, J.R.; Kim, S.C.; Kim, B.J. Effects of Gamisoyo-San Decoction, a Traditional Chinese Medicine, on Gastrointestinal Motility. Digestion. 2018, 98, 231-237.
42.Kim, J.N.; Nam, J.H.; Lee, J.R.; Kim, S.C.; Kim, B.J. The Traditional Medicine Banhasasim-Tang Depolarizes Pacemaker Potentials of Cultured Interstitial Cells of Cajal through M3 Muscarinic and 5-HT3 Receptors in Murine Small Intestine. Digestion 2020, 101, 536-551.
48.Lee, H.T.; Seo, E.K.; Chung, S.J.; Shim, C.K. Effect of an aqueous extract of dried immature fruit of Poncirus trifoliate (L.) Raf. on intestinal transit in rodents with experimental gastrointestinal motility dysfunctions. J. Ethnopharmacol. 2005, 102, 302–306.
49.Lee, M.C.; Ha, W.; Park, J.; Kim, J.; Jung, Y.; Kim, B.J. Effects of Lizhong Tang on gastrointestinal motility in mice. World J. Gastroenterol. 2016, 22, 7778-7786.
50.Kim, H.J.; Han, T.; Kim, Y.T.; So, I.; Kim, B.J. Magnolia Officinalis Bark Extract Induces Depolarization of Pacemaker Potentials Through M2 and M3 Muscarinic Receptors in Cultured Murine Small Intestine Interstitial Cells of Cajal. Cell. Physiol. Biochem. 2017, 43, 1790-1802.
51.Han, D.; Kim, J.N.; Kwon, M.J.; Han, T.; Kim, Y.T.; Kim, B.J. Salvia miltiorrhiza induces depolarization of pacemaker potentials in murine small intestinal interstitial cells of Cajal via extracellular Ca2+ and Na+ influx. Mol Med Rep. 2021, 23, 348.
52.Moon, S.B.; Choi, N.R.; Kim, J.N.; Kwon, M.J.; Kim, B.S.; Ha, K.T.; Lim, E.Y.; Kim, Y.T.; Kim, B.J. Effects of black garlic on the pacemaker potentials of interstitial cells of Cajal in murine small intestine in vitro and on gastrointestinal motility in vivo. Anim Cells Syst (Seoul). 2022, 26, 37-44.
53.Nair, A.B.; Jacob, S. A simple practice guide for dose conversion between animals and humans. J. Basic Clin. Pharm. 2016, 7, 27-31
54.Zou, W.; Zhou, H.; Hu, J.; Zhang, L.; Tang, Q.; Wen, X.; Xiao, Z.; Wang, W. Rhizoma Smilacis Glabrae inhibits pathogen-induced upper genital tract inflammation in rats through suppression of NF-κB pathway. J. Ethnopharmacol. 2017, 202, 103-113.
6) In the material and methods section, please describe how many mice were in each experimental group. Please, elaborate on the description of each group and what treatments were received including, each group name (normal, control, 0.01, 0.1, and 1 g/kg), the dose received, route for administration, and duration of administration of the extract.
Please, also comment on the adequacy of the sample size calculation. How did the authors decide on using the listed number of animals per experimental group? Authors are advised to address this point and add the answers/proper citations to a separate section named as the animal experimental design.
Responses) We added the answers in animal experiments in the material and methods section.
A total of 48 mice (Control (n=12, male 6: female 6), SGR 1 mg (n=12, 6:6), SGR 3 mg (n=12, 6:6), SGR 5 mg (n=12, 6:6)); 4‑8 days old; weighing 2.0‑2.3 g) of the Institute of Cancer Research (ICR) mice from the Samtako Bio Korea Co., Ltd. (Osan, Republic of Korea) were used for the ICC experiments, 86 male mice (Naïve (n=15), Control (n=15), SGR 0.01 mg (n=14), SGR 0.1 mg (n=15), SGR 1 mg (n=14), Mosapride (n=13)); 7 weeks old; weighing 23‑26 g) for the gastric emptying experiments. Also, 59 male mice (Naive (n=15), SGR 0.01 mg (n=15), SGR 0.1 mg (n=14), SGR 1 mg (n=15)); 7 weeks old; weighing 23‑26 g) for the normal ITR experiments and 74 male mice (Naive (n=15), Control (n=15); SGR 0.01 mg (n=15), SGR 0.1 mg (n=15), SGR 1 mg (n=14); 7 weeks old; weighing 23‑26 g) for the ITR experiments with GMD were used. All mice were housed in a specific pathogen‑free laboratory environment under a controlled temperature (20±2ËšC) and humidity (49±5%) with day and night cycles (light on at 7:00 a.m. and light off at 7:00 p.m) and ad libitum access to normal diet and autoclaved water. During the study, indicators of the general condition of the mice were observed daily, such as fur brightness, food and water intake, defecation and behavior. In the ICC experiments, ICC were placed under a microscope after making the cells, and SGR were administered to the cells to confirm the change of pacemaker potentials, and in the gastric emptying and ITR experiments, SGR were administered directly to the mouse mouth. We have been conducting ICC and animal experiments in the gastrointestinal tract for more than 20 years. It is well made without difficulty in culturing ICC and creating a gastrointestinal pathological animal model. Until now, gastrointestinal experiments were generally conducted with 10-15 animals, and sufficient good results and reproducibility could be confirmed. In this paper, about 10-15 animals were used.
7) To avoid readers’ confusion, the current animal study design should better be presented in a separate figure in the material and methods section.
Responses) In order to prevent confusion among readers, we newly created and organized animal experiments, and reorganized the other parts for each experiment.
8) Section 4.2.4. and 4.2.5. should be completely rewritten again. All the details of the animal groups, treatments, etc. must be detailed. It is not sufficient to just add the references; details of the experimental work should be described in adequate detail.
Responses) We re-checked and re-written the experimental method.
4.2.4. Gastric emptying was assessed by administering a 0.05% (w/v) phenol red solution (0.5 mL/mouse) 30 min after administering SGR. The mice were euthanized 30 min after treatment with phenol red, their stomachs were removed immediately, and the weights were measured. Next, the stomach was treated with 5 mL of 0.1N sodium hydroxide solution to check the optical density of the phenol red remaining in the stomach; 0.5 mL of trichloroacetic acid (20% w/v). The homogenate was centrifuged at 3000 rpm for 20 min. One milliliter of the supernatant was added to 4 mL of 0.5N sodium hydroxide solution, and the optical density of this pink liquid was measured using a spectrophotometer (560 nm). For experiments, mice were fasted for 19 h with a free supply of tap water. The selection of the phenol red solution volume (500 µL) and the 50% delayed gastric emptying time point (30 min after intraperitoneal injection of 10 mg/kg of loperamide) was done according to earlier study protocols was referenced [55,56]. The above emission values were obtained according to the formula: Gastric emptying (%) = (1-X/Y) *100. X: Optical density of the phenol red remaining on it. Y: Optical density of the phenol red mixture with sodium hydroxide under test tube conditions.
4.2.5. Evans blue was orally administered after SGR administration, and animals were sacrificed 30 minutes later. And ITR was measured by checking the length that Evans Blue passed in the intestine. The ITR was measured as a percentage of the total length passed by Evans Blue. A peritoneal irritation by acetic acid (PIA) model, one of the experimental GI motility dysfunction models, was used. Peritoneal irritation was induced using acetic acid (AA) in mice 30 min after the intragastric administration of SGR. PIA was induced by an intraperitoneal (i.p.) injection of acetic acid (0.6%, w/v, in saline) at a dose of 10 ml/kg, as previously described [37,39]. After injecting acetic acid, mice were placed in individual cages and allowed to recover for 30 min.
9) Since the study involves several experimental groups/treatments, statistical analysis is typically analyzed by ANOVA followed by a post-hoc test e.g., Tukey-Kramer. Please, add the name of the post-hoc test used by the authors.
Responses) We added Bonferroni, the name of the post-hoc test.
We analyzed the results with ANOVA with Bonferroni post-hoc tests for multiple comparisons
10) In the statistical analysis section, did the authors check data normality and homogeneity before proceeding to one-way ANOVA? Authors are advised to address this point and add the answers to the comment in the material and methods section.
Responses) We checked the normality and homogeneity before proceeding to one-way ANOVA.
11) To make all figure legends stand-alone, authors are advised to add the full name of the used abbreviations at the end of each legend including SGR, GMD, ICC, etc. Authors are advised to address this point and add the answers to the relevant figure legends.
Responses) We added the full name of the used abbreviations at the end of each legend.
12) In Figures 6-8, the figure captions must include information on the number of animals/replicates from which data were extracted. Authors are advised to address this point and add the answers to the relevant figure legends.
Responses) We included information on the number of animals in the relevant figure legends.
13) In the discussion section, authors are advised to describe the reported adverse effects of the extract from the literature.
Responses) We described the reported adverse effects of the extract from the literature in the discussion section.
In general, traditional medicine herbal medicines are known to be contaminated with heavy metals, microorganisms, pesticides, etc. and can cause serious side effects [21]. In addition, if herbal medicines change in a similar shape, serious complications occur [21]. Side effects caused by SGR are not well known and are known to cause hepatotoxicity, a common side effect of herbal medicines [22]. In addition, the Chinese classic book shows that LD50 in mice is 161 g/kg and 100 g/kg in Rat, and 45 consecutive days of administration at this concentration resulted in significant experimental animal death (P<0.01), and Rat's BUN exceeded the normal range [23]. In addition, when administered at 50 g/kg to Rat for 60 days, abnormalities in BUN occurred partially, but recovered after discontinuation of administration, and it was reported that degenerative necrosis appeared in hepatocytes and renal cells [23]. Therefore, in this paper, when administered at a maximum SGR of 5 g/kg for 14 days, there was no special change in important organs, so it is considered that there was no toxic reaction.
21.Xu, B.; Cheng, Q.; So, W.K.W. Review of the Effects and Safety of Traditional Chinese Medicine in the Treatment of Cancer Cachexia. Asia Pac. J. Oncol. Nurs. 2021, 8, 471-486.
22.Cheng, S.; Sun, H.; Li. X.; Yan, J.; Peng, Z.; You, Y.; Zhang, L.; Chen, J. Effects of Alismatis Rhizoma and Rhizoma Smilacis Glabrae Decoction on Hyperuricemia in Rats. Evid. Based Complement. Alternat. Med. 2019, 2019, 4541609.
23.Zhang, M.Z.; Dong, Z.H.; She, J. Modern Study Of Traditional Chinese Medicine, Vol. 1.; Xueyuan Press: Beijing, China, 1997; p.316.
14) The manuscript needs to be carefully checked by a native English speaker for grammar and typos. Some typos/syntax errors are present in the manuscript which need to be addressed, for example:
- In line 67, the authors state that “Network pharmacology analysis was conducted according to the schematic in Figure 1”.
Please, consider correcting the above statement to “Network pharmacology analysis was conducted according to the scheme in Figure 1”.
Responses) We carefully checked by a native English speaker.

Round 2
Reviewer 1 Report
As the authors addressed the reviewers' comments, I suggest acceptance of the manuscript.